# MULTI-STAGE INFLUENCE FUNCTION

## ABSTRACT

Multi-stage training and knowledge transfer from a large-scale pretraining task to various finetuning tasks have revolutionized natural language processing (NLP) and computer vision (CV), with state-of-the-art performances constantly being improved. In this paper, we develop a multi-stage influence function score to track predictions from a finetuned model all the way back to the pretraining data. With this score, we can identify the pretraining examples in the pretraining task that contribute most to a prediction in the finetuning task. The proposed multi-stage influence function generalizes the original influence function for a single model in Koh & Liang (2017), thereby enabling influence computation through both pretrained and finetuned models. We study two different scenarios with the pretrained embeddings fixed or updated in the finetuning tasks. We test our proposed method in various experiments to show its effectiveness and potential applications.

## 1 INTRODUCTION

Multi-stage training has become increasingly important and has achieved state-of-the-art results in many tasks. In NLP applications, it is now a common practice to first learn word embeddings (e.g., word2vec (Mikolov et al., 2013), GloVe (Pennington et al., 2014)) or contextual representations (e.g., ELMo (Peters et al., 2018), BERT (Devlin et al., 2018)) from a large unsupervised corpus, and then refine or finetune the model on supervised end tasks. In computer vision applications, it is common to use a pretrained CNN as a feature extractor and only finetune top-layer networks through training on the end task. Also, it has been demonstrated that pretraining ResNet (He et al., 2016) with large hashtag data can greatly benefit many end tasks (Mahajan et al., 2018). Intuitively, the successes of these multi-stage learning paradigms are due to knowledge transfer from pretraining tasks to the end task. However, current approaches using multi-stage learning are usually based on trial-and-error and many fundamental questions remain unanswered. For example, which part of the pretraining data/task contributes most to the end task? How can one detect "false transfer" where some pretraining data/task could be harmful for the end task? If a testing point is wrongly predicted by the finetuned model, can we trace back to the problematic examples in the pretraining data? Answering these questions requires a quantitative measurement of how the data and loss function in the pretraining stage influence the end model, which has not been studied in the past and will be the main focus of this paper.

To find the most influential training data responsible for a model's prediction, the influence function was first introduced by Cook & Weisberg (1980) from a robust statistics point of view. Recently, as large-scale applications become more challenging for influence function computation, Koh & Liang (2017) proposed to use a first-order approximation to measure the effect of removing one training point on the model's prediction, to overcome computational challenges. More broadly, there are many works using influence functions to investigate the impact of training data on models in various machine learning applications, such as tracing back the origins of bias in the word embeddings generated by GloVe (Brunet et al., 2019), and understanding and mitigating disparate impact to improve model fairness (Wang et al., 2019). However, all of the existing influence score computation algorithms study the case of single-stage training – where there is only one model with one set of training/prediction data in the training process. To the best of our knowledge, the influence of pretraining data on a subsequent finetuning task and model has not been studied, and it is nontrivial to apply the original influence function in (Koh & Liang, 2017) to this scenario.

In this work, we derive the influence function from pretraining data to the end task in multi-stage training. Since the computation involves several expensive Hessian vector products, we also show how to compute the influence function efficiently in large-scale problems. Based on this technique, we show that

- in real datasets and experiments across various vision and NLP tasks, predictions using the technique and actual influence for the pretraining data to the finetuned model are highly correlated (Pearson's $r$ score to be around 0.6). This shows the effectiveness of our proposed technique for computing the influence scores in multi-stage models;
- the influence for the pretraining data to the finetuned model can be split into two parts: the influence of the pretraining data on the pretrained model, and influence of the pretraining data on the finetuned model. Therefore the testing data from the finetuning task will be impacted by changes in the pretraining data, which can be quantified using our proposed technique;
- the influence of the pretraining data on the finetuning task is highly dependent on 1) the similarity of two tasks or stages, 2) and the number of training data in the finetuning task. Thus our proposed technique provides a novel way to measure how the pretraining data helps or benefits the finetuning data.

## 2 RELATED WORK

Multi-stage model training that trains models in many stages on different tasks to improve the end-task has been used widely in many machine learning areas, such as in transfer learning (Ando & Zhang, 2005) and zero-shot learning (Larochelle et al., 2008). Recently, multi-stage model training has achieved state-of-the-art performance by learning large embeddings or data representation as the pretraining step on a very large pretraining dataset, which is then followed by a finetune step with further training on the end-task. Examples include the recently proposed BERT (Devlin et al., 2018), which learns contextual embeddings on a large corpus with the pretraining tasks chosen to be predicting the masked words in a sentence and predicting whether one sentence is after another sentence. This contextual embedding is then used in finetuning tasks, such as a question answering task. ELMo (Peters et al., 2018) is widely used in multi-stage model training as a sentence feature extractor to benefit the end-task. Similarly, there are some works in computer vision that train an image representation model on a large number of images as the pretraining step, and then use the resulting features to finetune another task, such as particular image classification tasks. For example, (Mahajan et al., 2018) uses ResNet in the pretraining step, and the finetuning task is based on hashtag data. The rationale for this multi-stage model is that the pretraining task can learn some common or latent representation which could benefit the end task.

Another related line of research is on understanding machine learning models. One category of research is to explain predictions with respect to model variables, and trace back the contribution of variables to the prediction. For example, Oramas et al. (2019) automatically detects internal features in the set of classes in the pretrained model that are relevant to interpreting the prediction, and shows for various vision tasks that the proposed scheme can produce detailed explanations based on the features that are relevant to the targeted classes. Guo et al. (2019) aims to interpret variable-wise hidden states in an LSTM model to quantify variable importance and variable-wise temporal importance in the model.

Closely related research has sought to connect model prediction and training data, and trace back the most influential training data that are most responsible for the model prediction. Among them, the influence function (Cook & Weisberg, 1980; Koh & Liang, 2017), which aims to model the prediction changes when training data is added/removed, has been shown to be effective in many applications. There is a series of work on influence functions, including investigating the influence of a group of data on the prediction (Koh et al., 2019), using influence functions to detect bias in word embeddings (Brunet et al., 2019), and using it in preventing data poisoning attacks (Steinhardt et al., 2017), etc. All of these works only consider a single stage training procedure, and it is not straightforward to apply the existing influence functions to multi-stage models. In this paper, we propose to analyze the influence of pretraining data on predictions in the subsequent finetuned model and end task.

## 3 ALGORITHMS

In this section, we detail the procedure of multi-stage training, show how to compute the influence score for the multi-stage training, and then discuss how to scale up the computation.

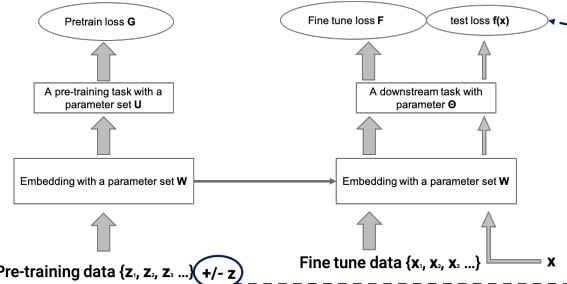

Figure 1: The setting for influence functions in multi-stage models. We consider a two-stage model, where we have a first pretrained model, and a second finetuned model for a desired end task. We seek to compute the influence of the pretraining data on predictions using testing data in the finetuning task.

### 3.1 MULTI-STAGE MODEL TRAINING

Multi-stage models, which train different models in consecutive stages, have been widely used in various ML tasks. Mathematically, let $\mathcal{Z}$ be the training set for pretraining task with data size $|\mathcal{Z}| = m$, and $\mathcal{X}$ be the training data for the finetuning task with data size $|\mathcal{X}| = n$. In pretraining stage, we assume the parameters of the pretrained network have two parts: the parameters $W$ that are shared with the end task, and the task-specific parameters $U$ that will only be used in the pretraining stage. Note that $W$ could be a word embedding matrix (e.g., in word2vec) or a representation extraction network (e.g., Elmo, BERT, ResNet), while $U$ is usually the last few layers that corresponds to the pretraining task. After training on the pretraining task, we obtain the optimal parameters $W^*, U^*$. The pretraining stage can be formulated as

$$\text{Pretrain Stage:} \quad W^*, \ U^* = \arg\min_{W,U} \frac{1}{m} \sum_{z \in \mathcal{Z}} g(z, \ W, \ U) := \arg\min_{W,U} G(W, U), \quad (1)$$

where $g(\cdot)$ is the loss function for the pretrain task.

In the finetuning stage, the network parameters are $W, \Theta$, where $W$ is shared with the pretraining task and $\Theta$ is the rest of the parameters specifically associated with the finetuning task. We will initialize the $W$ part by $W^*$. Let $f(\cdot)$ denote the finetuning loss, there are two cases when finetuning the end-task:

- Finetuning Case 1: Fixing embedding parameters $W = W^*$, and only finetune $\Theta$:

$$\Theta^* = \arg\min_{\Theta} \frac{1}{n} \sum_{x \in \mathcal{X}} f(x, \ W^*, \ \Theta) := \arg\min_{\Theta} F(W^*, \Theta). \quad (2)$$

- Finetuning Case 2: finetune both the embedding parameters $W$ (initialized from $W^*$) and $\Theta$. Sometimes updating the embedding parameters $W$ in the finetuning stage is necessary, as the embedding parameters from the pretrained model may not be good enough for the finetuning task. This corresponds to the following formulation:

$$W^{**}, \Theta^* = \arg\min_{W,\Theta} \frac{1}{n} \sum_{x \in \mathcal{X}} f(x, \ W, \ \Theta) := \arg\min_{W,\Theta} F(W, \Theta). \quad (3)$$

### 3.2 INFLUENCE FUNCTION FOR MULTI-STAGE MODELS

We derive the influence function for the multi-stage model to trace the influence of pretraining data on the finetuned model. In Figure 1 we show the task we are interested in solving in this paper. Note

that we use the same definition of influence function as (Koh & Liang, 2017) and discuss how to compute it in the multi-stage training scenario. As discussed at the end of Section 3.1, depending on whether or not we are updating the shared parameters $W$ in the finetuning stage, we will derive the influence functions under two different scenarios.

### 3.2.1 CASE 1: EMBEDDING PARAMETERS $W$ ARE FIXED IN FINETUNING

To compute the influence of pretraining data on the finetuning task, the main idea is to perturb one data example in the pretraining data, and study how that impacts the test data. Mathematically, if we perturb a pretraining data example $z$ with loss change by a small $\epsilon$, the perturbed model can be defined as

$$\hat{W}_\epsilon, \hat{U}_\epsilon = \arg\min_{W,U} G(W, U) + \epsilon g(z, W, U). \tag{4}$$

Note that choices of $\epsilon$ can result in different effects in the loss function from the original solution in (1). For instance, if we set $\epsilon = -\frac{1}{n}$, then we are removing the training data $z$ in the pretraining dataset.

For the finetuning stage, since we consider Case 1 where the embedding parameters $W$ are fixed in the finetuning stage, the new model for the end-task or finetuning task will thus be

$$\hat{\Theta}_\epsilon = \arg\min_{\Theta} F(\hat{W}_\epsilon, \Theta). \tag{5}$$

The influence function that measures the impact of a small $\epsilon$ perturbation on $z$ to the finetuning loss on a test sample $x_t$ from finetuning task is defined as

$$I_{z,x_t} := \frac{\partial f(x_t, \hat{W}_\epsilon, \hat{\Theta}_\epsilon)}{\partial \epsilon}\Big|_{\epsilon=0} \tag{6}$$

$$= \nabla_\Theta f(x_t, W^*, \Theta^*)^T \cdot I_{z,\Theta} + \nabla_W f(x_t, W^*, \Theta^*)^T \cdot I_{z,W}. \tag{7}$$

$$with \ \ I_{z,\Theta} := \frac{\partial \hat{\Theta}_\epsilon}{\partial \epsilon}\Big|_{\epsilon=0} \ \ and \ \ I_{z,W} := \frac{\partial \hat{W}_\epsilon}{\partial \epsilon}\Big|_{\epsilon=0}, \tag{8}$$

where $I_{z,\Theta}$ measures the influence of $z$ on the finetuning task parameters $\Theta$, and $I_{z,W}$ measures how $z$ influences the pretrained model $W$. Therefore we can split the influence of $z$ on the test sample into two pieces: one is the impact of $z$ on the pretrained model $I_{z,W}$, and the other is the impact of $z$ on the finetuned model $I_{z,\Theta}$. It is worth mentioning that, due to linearity, if we want to estimate a set of test example influence function scores with respect to a set of pretraining examples, we can simply sum up the pair-wise influence functions, and so define

$$I_{\{z^{(i)}\},\{x_t^{(j)}\}} := \sum_i \sum_j I_{z^{(i)}, x_t^{(j)}}. \tag{9}$$

where $\{z^{(i)}\}$ and $\{x_t^{(j)}\}$ contain a set of pretraining data and finetuning test data. Next we will derive these two influence scores $I_{z,\Theta}$ and $I_{z,W}$ (see the detailed derivations in the appendix) in Theorem 1 below.

**Theorem 1.** *For the two-stage training procedure in* (1) *and* (2)*, we have*

$$I_{z,W} := \frac{\partial \hat{W}_\epsilon}{\partial \epsilon}\Big|_{\epsilon=0} = -\left[\left(\frac{\partial^2 G(W^*, U^*)}{\partial (W, U)^2}\right)^{-1}\left(\frac{\partial g(z, W^*, U^*)}{\partial (W, U)}\right)\right]_W \tag{10}$$

$$I_{z,\Theta} := \frac{\partial \hat{\Theta}_\epsilon}{\partial \epsilon}\Big|_{\epsilon=0} = \left(\frac{\partial^2 F(W^*, \Theta^*)}{\partial \Theta^2}\right)^{-1} \cdot \left(\frac{\partial^2 F(W^*, \Theta^*)}{\partial \Theta \partial W}\right) \cdot \left[\left(\frac{\partial^2 G(W^*, U^*)}{\partial (W, U)^2}\right)^{-1}\left(\frac{\partial g(z, W^*, U^*)}{\partial (W, U)}\right)\right]_W$$

*where $[\cdot]_W$ means taking the $W$ part of the vector.*

By plugging (10) into (7), we finally obtain the influence score of pretraining data $z$ on the finetuning task testing point $x_t$, $I_{z,x_t}$ as

$$I_{z,x_t} = \left[-\frac{\partial f(x, W^*, \Theta^*)^T}{\partial \Theta} \cdot \left(\frac{\partial^2 F(W^*, \Theta^*)}{\partial \Theta^2}\right)^{-1} \cdot \frac{\partial^2 F(W^*, \Theta^*)}{\partial \Theta \partial W} + \frac{\partial f(x, W^*, \Theta^*)^T}{\partial W}\right] I_{z,W} \tag{11}$$

Algorithm 1 shows how to compute the influence score in (11). The pseudocode for computing the influence function in (11) is shown in Algorithm 1.

---

**Algorithm 1:** Multi-Stage Influence Score with Fixed Embedding

---

1 **Input**: pretrain and finetune models with $W^*$, $\Theta^*$, and $U^*$; pretrain and finetune training data $\mathcal{Z}$ and $\mathcal{X}$; test example $x_t$; and a pretrain training example $z$;

2 **Output**: Influence function value $I_{z,x_t}$;

3 Compute fintune model's gradients $\frac{\partial f(x_t,W^*,\Theta^*)}{\partial \Theta}$ and $\frac{\partial f(x_t,W^*,\Theta^*)}{\partial W}$;

4 Compute the first inverse Hessian vector product $V_{ihvp1}(x_t) := (\frac{\partial^2 F(W^*,\Theta^*)}{\partial \Theta^2})^{-1}\frac{\partial f(x_t,W^*,\Theta^*)}{\partial \Theta}$;

5 Compute finetune loss's gradient w.r.t $W$: $\frac{\partial f(x_t,W^*,\Theta^*)^T}{\partial W} = V_{ihvp1}^T\frac{\partial^2 F(W^*\Theta^*)}{\partial \Theta \partial W} - \frac{f(x_t,W^*,\Theta^*)}{\partial W}$ and concatenate it with 0 to make it the same dimension as $(W,\ U)$;

6 Compute and save the second inverse Hessian vector product

$V_{ihvp2}^T(x_t) := [\frac{\partial f(x_t,\Theta^*,W^*)^T}{\partial W},\ 0](\frac{\partial^2 G(W^*,U^*)}{\partial(W,U)^2})^{-1}$ ;

7 Compute influence function score $I_{z,x_t} = V_{ihvp2}^T(x_t)\frac{\partial g(z,W^*,U^*)}{\partial(W,U)}$;

---

### 3.2.2 CASE 2: EMBEDDING PARAMETER $W$ IS ALSO UPDATED IN THE FINETUNING STAGE

For the second finetuning stage case in (3), we will also further train the embedding parameter $W$ from the pretraining stage. When $W$ is also updated in the finetuning stage, it is challenging to characterize the influence since the pretrained embedding $W^*$ is only used as an initialization. In general, the final model $(W^{**},\Theta^*)$ may be totally unrelated to $W^*$; for instance, when the objective function is strongly convex, any initialization of $W$ in (3) will converge to the same solution.

However, in practice the initialization of $W$ will strongly influence the finetuning stage in deep learning, since the finetuning objective is usually highly non-convex and initializing $W$ with $W^*$ will converge to a local minimum near $W^*$. Therefore, we propose to approximate the whole training procedure as

$$\bar{W},\bar{U} = \arg\min_{W,U} G(W,U) \tag{12}$$
$$W^*,\Theta^* = \arg\min_{W,\Theta}\{\alpha\|W - \bar{W}\|_F^2 + F(W,\Theta)\},$$

where $\bar{W},\bar{U}$ are optimal for the pretraining stage, $W^*,\Theta^*$ are optimal for the finetuning stage, and $0 \leq \alpha \ll 1$ is a small value. This is to characterize that in the finetuning stage, we are targeting to find a solution that minimizes $F(W,\Theta)$ and is close to $\bar{W}$. In this way, the pretrained parameters are connected with finetuning task and thus influence of pretraining data to the finetuning task can be tractable. The results in our experiments show that with this approximation, the computed influence score can still reflect the real influence quite well.

Similarly we can have $\frac{\partial\hat{\Theta}_\epsilon}{\partial\epsilon}$, $\frac{\partial\hat{W}_\epsilon}{\partial\epsilon}$, and $\frac{\partial\bar{W}_\epsilon}{\partial\epsilon}$ to measure the difference between their original optimal solutions in (12) and the optimal solutions from $\epsilon$ perturbation over the pretraining data $z$. Similar to (7), the influence function $I_{z,x_t}$ that measures the influence of $\epsilon$ perturbation to pretraining data $z$ on test sample $x_t$'s loss is

$$I_{z,x_t} := \frac{\partial f(x_t,\hat{W}_\epsilon,\hat{\Theta}_\epsilon)}{\partial\epsilon}\Big|_{\epsilon=0} = \frac{\partial f(x_t,W^*,\Theta^*)}{\partial(W,\Theta)}^T \left[\begin{array}{c}\frac{\partial\hat{W}_\epsilon}{\partial\epsilon}\big|_{\epsilon=0} \\ \frac{\partial\hat{\Theta}_\epsilon}{\partial\epsilon}\big|_{\epsilon=0}\end{array}\right]. \tag{13}$$

The influence function of small perturbation of $G(W,U)$ to $\bar{W},W^*,\Theta^*$ can be computed following the same approach in Subsection 3.2.1 by replacing $\bar{W}$ for $W^*$ and $[\Theta^*,W^*]$ for $\Theta^*$ in (10). This will lead to

$$\frac{\partial\bar{W}_\epsilon}{\partial\epsilon}\Big|_{\epsilon=0} = -\left[(\frac{\partial^2 G(\bar{W},\bar{U})}{\partial(W,U)^2})^{-1}(\frac{\partial g(z,\bar{W},\bar{U})}{\partial(W,U)})\right]_W \tag{14}$$

$$\left[\begin{array}{c}\frac{\partial\hat{\Theta}_\epsilon}{\partial\epsilon}\big|_{\epsilon=0} \\ \frac{\partial\hat{W}_\epsilon}{\partial\epsilon}\big|_{\epsilon=0}\end{array}\right] = \left[\begin{array}{cc}\frac{\partial^2 F(W^*,\Theta^*)}{\partial\Theta^2} & \frac{\partial^2 F(W^*,\Theta^*)}{\partial\Theta\partial W} \\ \frac{\partial^2 F(W^*,\Theta^*)}{\partial\Theta\partial W} & \frac{\partial^2 F(W^*,\Theta^*)}{\partial W^2} + 2\alpha I\end{array}\right]^{-1}\left[\begin{array}{c}0 \\ -2\alpha I\end{array}\right]\left[(\frac{\partial^2 G(\bar{W},\bar{U})}{\partial(W,U)^2})^{-1}(\frac{\partial g(z,\bar{W},\bar{U})}{\partial(W,U)})\right]_W$$

After plugging (14) into (13), we will have the influence function $I_{z,x_t}$. Similarly, the algorithm for computing $I_{z,x_t}$ for Case 2 can follow Algorithm 1 for Case 1 by replacing gradient computation.

### 3.3 IMPLEMENTATION DETAILS

The influence function computation for multi-stage model is presented in the previous section. As we can see in Algorithm 1 that the influence score computation involves many Hessian matrix operations, which will be very expensive and sometimes unstable for large-scale models. We used several strategies to speed up the computation and make the scores more stable.

**Large Hessian Matrices**  A Hessian matrix $H$ has a size of $p \times p$, where $p$ is the number of parameters in the model. For large deep learning models with thousands or even millions of parameters, it is almost impossible to fit a $p \times p$ Hessian into memory. Also, to invert a Hessian requires $O(p^3)$ operations. Similar to Koh & Liang (2017), we avoid explicitly computing and storing the Hessian matrix and its inverse, and instead compute product of the inverse Hessian with a vector directly. Every time we need an inverse Hessian vector product $v = H^{-1}b$, we invoke conjugate gradients (CG), which transforms the linear system problem into an quadratic optimization problem $H^{-1}b \equiv \arg\min_x\{\frac{1}{2}x^T H x - b^T x\}$. In each iteration of CG, instead of computing $H^{-1}b$ directly, we will compute a Hessian vector product, which can be efficiently done by backprop through the model twice with $O(p)$ time complexity. The aforementioned conjugate gradient method requires the Hessian matrix to be positive definite. However, in practice the Hessian may have negative eigenvalues, since we run a SGD and the final $H$ may not at a local minimum exactly. To tackle this issue, we solve

$$\arg\min_x\{\frac{1}{2}x^T H^2 x - b^T H x\}, \tag{15}$$

whose solution can be shown the same as $\arg\min_x\{\frac{1}{2}x^T H x - b^T x\}$ since the Hessian matrix is symmetric. $H^2$ is guaranteed to be positive definite as long as $H$ is invertible, even when $H$ has negative eigenvalues. If $H^2$ is not ill-conditioned, we can solve (15) directly. The rate of convergence of CG depends on $\frac{\sqrt{\kappa(H^2)}-1}{\sqrt{\kappa(H^2)}+1}$, where $\kappa(H^2)$ is the condition number of $H^2$, which can be very large if $H^2$ is ill-conditioned. When $H^2$ is ill-conditioned, to stabilize the solution and to encourage faster convergence, we add a small damping term $\lambda$ on the diagonal and solve $\arg\min_x\{\frac{1}{2}x^T(H^2 + \lambda I)x - b^T H x\}$.

**Time Complexity**  As mentioned above, we can get an inverse Hessian vector product in $O(p)$ time. We assume there are $p_1$ parameters in our pretrained model and $p_2$ parameters in our finetuned model. Since $F$ and $G$ are summation of loss with respect to all pretraining or finetuning examples, it takes $O(mp_1)$ or $O(np_2)$ to compute a Hessian vector product, where $m$ is the number of pretraining examples and $n$ is the number of finetuning examples. We may also subsample the pretraining examples to estimate $G(W, U)$ when the number of pretraining examples is gigantic such as pretraining using One-Billion-word dataset (Chelba et al., 2013) for ELMo etc. For the two inverse Hessian vector products, the time complexity is $O(np_2 r)$ and $O(mp_1 r)$, where $r$ is the number of iterations in CG. For other operations in computing the influence score, vector product has a time complexity of $O(p_1)$ or $O(p_1)$, and computing the gradients of all pretraining examples has a complexity of $O(mp_1)$. So computing the total time complexity of computing a multi-stage influence score is $O((mp_1 + np_2)r)$

## 4 EXPERIMENTS

In this section, we will conduct experiment with real datasets on both vision and NLP tasks to show the effectiveness of our proposed influence function. We will show the results in the main text on the vision tasks, and some qualitative results on NLP task related to ELMo are presented in Section C in Appendix.

### 4.1 INFLUENCE FUNCTION CORRELATION WITH REAL SCORE

To show that our proposed influence score are a good approximation, we evaluate our proposed multi-stage influence function on two CNN models with CIFAR and MNIST datasets. The model structures are shown in Table A in Appendix. We use Tanh for all activations. For both MNIST and CIFAR models, CNN layers are used as embeddings and fully connected layers are task-specific. At the pretraining stage, we train the models with examples from only two classes ("bird" vs. "frog") for

CIFAR and four classes (0, 1, 2, and 3) for MNIST. The resulting embedding is used in the finetuning tasks, where we finetune the model with the examples from the remaining eight classes in CIFAR or the other 6 numbers in MNIST task for classification. In order to make the experiments closer to typical real finetuning situations, we reduce the size of the training set in the finetuning task by subsampling.

In this experiment we test the correlation between individual pretraining example's multi-stage influence function and the real loss difference when the pretraining examples are removed. We test two cases (as mentioned in Section 3.1) – where the pretrained embedding is fixed, and where it is updated during finetuning. For both MNIST and CIFAR, we first train the embedding with the binary classification for $T_{\text{pretrain}}$ steps. The embedding is then used in the finetuned model. Then we train the finetuned model for $T_{\text{finetune}}$ steps. Depending on different scenarios which will be explained below, the embedding may be fixed or updated in the finetuning task training. For a given example in the pretraining data, we calculate its influence function score with respect to each test example in the finetuning task test set using the method presented in Section 3. To evaluate this pretraining example's contribution to the overall performance of the model, we sum up the influence function scores across the whole test set in the finetuning task.

To validate the score, we remove that pretraining example and go through the aforementioned process again by updating the model. In the updating process we further train the pretrained and finetuned models for $T'_{\text{pretrain}}$ and $T'_{\text{finetune}}$ steps from the original model checkpoints. Note that in this process the pretraining is conducted with the new leave-one-out pretraining training set, while the training set for the finetuning task is intact. Due to computation constraints, we only use the top 100 pretraining examples with the largest influence function absolute values in this experiments to get 100 score-difference pairs. Then we run a linear regression between the true loss difference values obtained and the influence score computed to show their correlation. The detailed hyperparameters used in these experiments are presented in Appendix B.

### 4.1.1 EMBEDDING IS FIXED

In Figure 2 we show the correlation results of CIFAR and MNIST models when the embedding is fixed in finetuning task training. Though we make many approximations in our formulation, from Figures 2a and 2b we can see that there is a clear linear correlation between the true loss difference and the influence function scores obtained. The correlation is evaluated with Pearson's $r$ value. This supports our argument that we can use this score to detect the examples in the pretraining set which contributes most to the model's performance. In Figure 3 we demonstrate the misclassified test

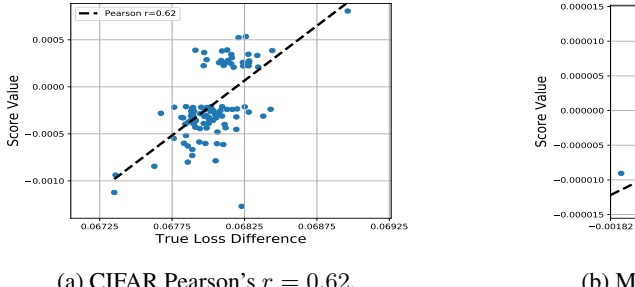

(a) CIFAR Pearson's $r = 0.62$.     (b) MNIST Pearson's $r = 0.47$.

Figure 2: CIFAR and MNIST model true loss difference vs. the influence function scores by (8). The loss is calculated as the sum of all test examples.

images in the finetuning task and the images with the largest positive influence function score in the pretraining dataset. Examples with large positive influence score are expected to have negative effect on the model's performance since intuitively when they are added to the pretraining dataset, the loss of the test example will increase. From Figure 3 we can indeed see that the identified examples are with low quality, and they can be easily misclassified even with human eyes.

One may doubt the effectiveness of the expensive inverse Hessian computation in our formulation. As a comparison, we replace all inverse Hessians in (11) with identity matrices to compute the

influence function score for the MNIST model. The results are shown in Figure 4 with a much smaller Pearson's $r$ of 0.17. This again shows effectiveness of our proposed influence function.

### 4.1.2 EMBEDDING IS UPDATED IN FINETUNE

Practically, the embedding can also be updated in the finetuning process. In Figure 5 we show the correlation between true loss difference and influence function score values using (13). We can see that even under this challenging condition, our multi-stage influence function from (13) still has a strong correlation with the true loss difference, with a Pearson's $r = 0.40$.

### 4.2 THE FINETUNING TASK'S SIMILARITY TO THE PRETRAINING TASK

In this experiment, we explore the relationship between influence function score and finetuning task similarity with the pretraining task. Specifically, we study whether the influence function score will increase in absolute value if the finetuning task is very similar to the pretraining task. To do this, we use the CIFAR embedding obtained from a "bird vs. frog" classification and test its influence function scores on two finetuning tasks. The finetuning task A is exactly the same as the pretraining "bird vs. frog" classification, while the finetuning task B is a classification on two other classes ("automobile vs. deer"). All hyperparameters used in the two finetuning tasks are the same. In Figure 6, for both tasks we plot the distribution of the influence function values with respect to each pretraining example. We sum up the influence score for all test examples. We can see that, the first finetuning task influence function has much larger absolute values than that of the second task. The average absolute value of task A influence function score is 0.055, much larger than that of task B, which is 0.025. This supports the argument that if pretraining task and finetuning task are similar, the pretraining data will have larger influence on the finetuning task performance.

### 4.3 INFLUENCE FUNCTION SCORE WITH DIFFERENT NUMBER OF FINETUNE EXAMPLES

We also study the relationship between the influence function scores and number of examples used in finetuning. In this experiment, we update the pretrained embedding in finetuning stage. We use the same pretraining and finetuning task as in Section 4.1. In Figure 7, model C is the model used in

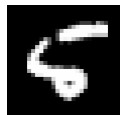 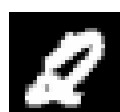 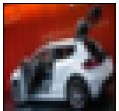 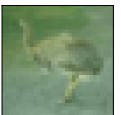

| test example
prediction=6
true label=5 | pretrain example
influence score=91.6
true label=2 | test example
prediction="cat"
true label="automobile" | pretrain example
influence score=1060.9
true label="bird" |

Figure 3: Identifying the pretrain example with largest influence function score which contributes an error in the finetune task.

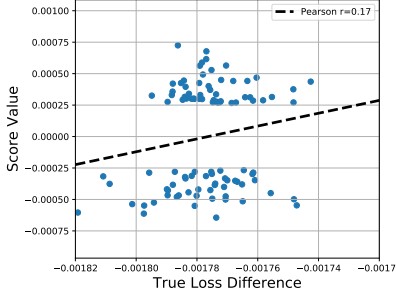 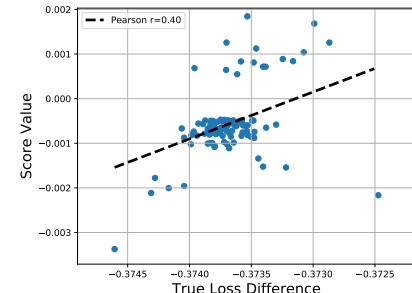

Figure 4: MNIST model true loss difference and influence function with all Hessians replaced by identity matrices. Pearson's $r = 0.17$.

Figure 5: CIFAR example using $\ell_2$ regularization formulation in (13) when embedding is not fixed in finetune. Pearson's $r = 0.40$.

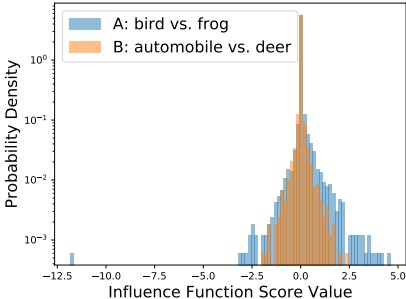 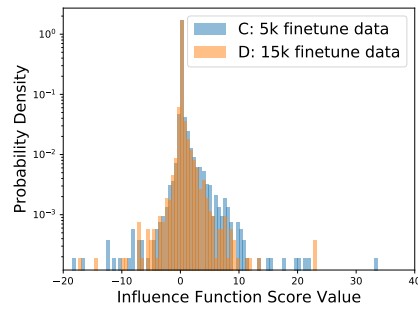

Figure 6: Two different finetuning task distribution of influence function scores with respect to each pretrain example. Each pretraining example's influence score value is summed up for all test examples. The pretrained embedding is fixed in finetuning. For both finetuning tasks, the pretrained model is the same, and is trained using "bird vs. frog" in CIFAR. For model A, finetuning task and pretraining task are the same. The average absolute values of influence function scores for models A and B are 0.055 and 0.025, respectively.

Figure 7: Two different finetuning task distribution of influence function scores which respect to each pretraining example. Each pretraining example's influence score value is summed up for all test examples. The pretrained embedding is also updated in finetuning. The pretraining and finetuning tasks are the same as in Section 4.1.2. Model D's number of finetuning examples and finetuning steps are 3X of model C's. The average absolute values of influence function scores for Models C and D are 0.22 and 0.15, respectively.

Section 4.1.2 while in model D we triple the number of finetuning examples as well as the number of finetuning steps. Figure 7 demonstrates the distribution of each pretraining examples' influence function score with the whole test set. The average absolute value of influence function score in model D is 0.15, much less than that of model C. This indicates that with more finetuning examples and more finetuning steps, the influence of pretraining data to the finetuning model's performance will decrease. This makes sense as if the finetuning data does not have sufficient information for training a good finetuning task, then pretraining data will have more impact on the finetuning task.

## 5   CONCLUSION

We introduce a multi-stage influence function to evaluate pretraining examples' contribution to finetuned model's prediction. Two different cases are studied: the pretrained embedding is fixed in finetuning or the pretrained embedding is updated in finetuning. We test our method on both CV and NLP tasks. Our experimental results show strong correlation between the proposed multi-stage influence function scores and the true loss difference when an example is removed from the pretraining data. We believe this is a promising way to connect finetuned model's performance with pretraining data directly.

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

## A    PROOF OF THEOREM 1

*Proof.* Since $\hat{\Theta}_\epsilon, \hat{U}_\epsilon, \hat{W}_\epsilon$ are optimal solutions, and thus satisfy the following optimality conditions:

$$0 = \frac{\partial}{\partial \Theta} F(\hat{W}_\epsilon, \hat{\Theta}_\epsilon) \tag{16}$$

$$0 = \frac{\partial}{\partial (W,U)} G(\hat{W}_\epsilon, \hat{U}_\epsilon) + \epsilon \frac{\partial}{\partial (W,U)} g(z, \hat{W}_\epsilon, \hat{U}_\epsilon), \tag{17}$$

where $\partial(W,U)$ means concatenate the $U$ and $W$ as $[W,U]$ and compute the gradient w.r.t $[W,U]$. We define the changes of parameters as $\Delta W_\epsilon = \hat{W}_\epsilon - \hat{W}$, $\Delta \Theta_\epsilon = \hat{\Theta}_\epsilon - \hat{\Theta}$, and $\Delta U_\epsilon = \hat{U}_\epsilon - \hat{U}$. Applying Taylor expansion to the rhs of (17) we get

$$0 \approx \frac{\partial}{\partial (W,U)} G(W^*, U^*) + \frac{\partial^2 G(W^*, U^*)}{\partial (W,U)^2} \begin{bmatrix} \Delta W_\epsilon \\ \Delta U_\epsilon \end{bmatrix} + \epsilon \frac{\partial g(z, W^*, U^*)}{\partial (W,U)} + \epsilon \frac{\partial^2 g(z, W^*, U^*)}{\partial (W,U)^2} \begin{bmatrix} \Delta W_\epsilon \\ \Delta U_\epsilon \end{bmatrix} \tag{18}$$

Since $W^*, U^*$ are optimal of unperturbed problem, $\frac{\partial}{\partial (W,U)} G(W^*, U^*) = 0$, and we have

$$\begin{bmatrix} \Delta W_\epsilon \\ \Delta U_\epsilon \end{bmatrix} \approx - \left( \frac{\partial^2 G(W^*, U^*)}{\partial (W,U)^2} + \epsilon \frac{\partial^2 g(z, W^*, U^*)}{\partial (W,U)^2} \right)^{-1} \left( \frac{\partial g(z, W^*, U^*)}{\partial (W,U)} \right) \epsilon \tag{19}$$

Since $\epsilon \to 0$, we have further approximation

$$\begin{bmatrix} \Delta W_\epsilon \\ \Delta U_\epsilon \end{bmatrix} \approx \left( \frac{\partial^2 G(W^*, U^*)}{\partial (W,U)^2} \right)^{-1} \left( \frac{\partial g(z, W^*, U^*)}{\partial (W,U)} \right) \epsilon \tag{20}$$

Similarly, based on (16) and applying first order Taylor expansion to its rhs we have

$$0 \approx \frac{\partial F(W^*, \Theta^*)}{\partial \Theta} + \frac{\partial^2 F(W^*, \Theta^*)}{\partial \Theta \partial W} \cdot \Delta W_\epsilon + \frac{\partial^2 F(W^*, \Theta^*)}{\partial \Theta^2} \Delta \Theta_\epsilon. \tag{21}$$

Combining (21) with (20) we have

$$\Delta \Theta_\epsilon \approx \left( \frac{\partial^2 F(W^*, \Theta^*)}{\partial \Theta^2} \right)^{-1} \cdot \left( \frac{\partial^2 F(W^*, \Theta^*)}{\partial \Theta \partial W} \right) \cdot \left[ \left( \frac{\partial^2 G(W^*, U^*)}{\partial (W,U)^2} \right)^{-1} \left( \frac{\partial g(z, W^*, U^*)}{\partial (W,U)} \right) \right]_W \epsilon$$

where $[\cdot]_W$ means taking the $W$ part of the vector. Therefore,

$$I_{z,W} := \frac{\partial \hat{W}_\epsilon}{\partial \epsilon} \Big|_{\epsilon=0} = - \left[ \left( \frac{\partial^2 G(W^*, U^*)}{\partial (W,U)^2} \right)^{-1} \left( \frac{\partial g(z, W^*, U^*)}{\partial (W,U)} \right) \right]_W \tag{22}$$

$$I_{z,\Theta} := \frac{\partial \hat{\Theta}_\epsilon}{\partial \epsilon} \Big|_{\epsilon=0} = \left( \frac{\partial^2 F(W^*, \Theta^*)}{\partial \Theta^2} \right)^{-1} \cdot \left( \frac{\partial^2 F(W^*, \Theta^*)}{\partial \Theta \partial W} \right) \cdot \left[ \left( \frac{\partial^2 G(W^*, U^*)}{\partial (W,U)^2} \right)^{-1} \left( \frac{\partial g(z, W^*, U^*)}{\partial (W,U)} \right) \right]_W \tag{23}$$

□

## B    MODELS AND HYPERPARAMETERS FOR THE EXPERIMENTS IN SECTIONS 4.1, 4.2 AND 4.3

The model structures we used in Sections 4.1, 4.2 and 4.3 are listed in Table A. As mentioned in the main text, for all models, CNN layers are used as embeddings and fully connected layers are task-specific. The number of neurons on the last fully connected layer is determined by the number of classes in the classification. There is no activation at the final output layer and all other activations are Tanh.

- For MNIST experiments in Section 4.1.1, we train a four-class classification (0, 1, 2, and 3) in pretraining. All examples in the original MNIST training set with with these four labels are used in pretraining. The finetuning task is to classify the rest six classes, and

we subsample only 5000 examples to finetune. The pretrained embedding is fixed in finetuning. We run Adam optimizer in both pretraining and finetuning with a batch size of 512. The pretrained and finetuned models are trained to converge. When validating the influence function score, we remove an example from pretraining dataset. Then we re-run the pretraining and finetuning process with this leave-one-out pretraining dataset starting from the original models' weights. In this process, we only run 100 steps for pretraining and finetuning as the models converge. When computing the influence function scores, the damping term for the pretrained and finetuned model's Hessians are $1 \times 10^{-2}$ and $1 \times 10^{-8}$, respectively. We sample 1000 pretraining examples when computing the pretraind model's Hessian summation.

- For CIFAR experiments in Section 4.1.1, we train a two-class classification ("bird" vs "frog") in pretraining. All examples in the original CIFAR training set with with these four labels are used in pretraining. The finetuning task is to classify the rest eight classes, and we subsample only 5000 examples to finetune. The pretrained embedding is fixed in finetuning. We run Adam optimizer to train both pretrained and finetuned model with a batch size of 128. The pretrained and finetuned models are trained to converge. When validating the influence function score, we remove an example from pretraining dataset. Then we re-run the pretraining and finetuning process with this leave-one-out pretraining dataset starting from the original models' weights. In this process, we only run 6000 steps for pretraining and 3000 steps for finetuning. When computing the influence function scores, the damping term for the pretrained and finetuned model's Hessians are $1 \times 10^{-8}$ and $1 \times 10^{-6}$, respectively. Same hyperparameters are used in experiments in Sections 4.2 and 4.3. We also use these hyperparameters in Section 4.1.2's experiments, except that the pretrained embedding is updated in finetuning and the number of finetuning steps is reduced to 1000 in validation. The $\alpha$ constant in Equation 14 is chosen as 0.01. We sample 1000 pretraining examples when computing the pretrained model's Hessian summation.

| Dataset | MNIST | CIFAR |
|---|---|---|
| Embedding | CONV 32 5×5+1
MAX-POOL 2×2 +2
CONV 64 5×5+1
MAX-POOL 2×2 +2 | CONV 32 3×3+1
CONV 64 4×4+1
MAX-POOL 2×2 +2
CONV 128 2×2+1
MAX-POOL 2×2 +2
CONV 128 2×2+1
MAX-POOL 2×2 +2 |
| Task specific | FC <# classes> | FC 1500
FC <# classes> |

Table A: Model Architectures. "CONV $k$ $w \times h + s$" represents a 2D convolutional layer with $k$ filters of size $w \times h$ using a stride of $s$ in both dimensions. "MAX-POOL $w \times h + s$" represents a 2D max pooling layer with kernel size $w \times h$ using a stride of $s$ in both dimensions. "FC n" = fully connected layer with $n$ outputs. All activation functions are Tanh and last fully connected layers do not have activation functions. The number of neurons on the last fully connected layer is determined by the number of classes in the task.

## C ELMO EXPERIMENT

In this section we show influence function score results with ELMo. The finetune task is a binary sentiment classification of twitter[1] and the ELMo model is pretrained on a one-billion-word dataset Chelba et al. (2013). For the finetuned model, we add a hidden layer with 64 neurons and an output layer to build the classifier. The activation function is Tanh. For simplicity, we use the original pretrained ELMo embedding and the embedding is fixed in finetuning. We randomly sample a subset of 1000 sentences from one-billion-word dataset. For a test sentence, we list the pretrain sentences with the largest and the smallest absolute influence function score values in one-billion-word dataset.

---

[1] https://datahack.analyticsvidhya.com/contest/linguipedia-codefest-natural-language-processing-1/#data_dictionary

| Test Sentence | Max abs. score | Sentence in Pretrain | Min abs. score | Sentence in Pretrain |
|---|---|---|---|---|
| *Finally a transparent silicon case Thanks to my uncle :) #yay #Sony #Xperia #S #sonyexperias. . .* | -0.0049 | *Prof Slobodchikoff details the experiments he has done to reveal the hidden structure of the prairie dog 's language within the BBC natural history programme " Prairie dogs , talk of the town , " broadcast as part of the Natural World documentary series .* | $6.74 \times 10^{-9}$ | *He will be there to help you . "* |
| *Bout to go shopping again listening to music #iphone #justme #music #likeforlike #followforfollow. . .* | 0.0014 | *We are seeing the first big systematic investment in dance .* | $-6.30 \times 10^{-9}$ | *The war seemed to energize her , and she began to hang out with the American journalists based in London .* |
| *Ha! Not heavy machinery but it does what I need it to. @Apple really dropped the ball with that design. #drinkyourhaterade* | 0.00052 | *He 's been on the move since his comeback victory over Juan Manuel Marquez more than two weeks ago , a brutally efficient boxing display that generated a staggering 1 million pay-per-view buys .* | $-2.25 \times 10^{-9}$ | *Taylor said the syndicated TV psychologist broached the idea of the show to Spears ' handlers , who eventually decided that such a show would be " detrimental " to the family .* |

Table B: The list of test sentences and pretraining sentences with the largest and the smallest absolute influence function score values in our subset of pretraining data. The subset consists of 1000 random sentences from one-billion-word, which is used to pretrained ELMo embedding.

