# OpenReview forum: "MULTI-STAGE INFLUENCE FUNCTION"
_ICLR.cc/2020/Conference — Reject_

### Official Review · AnonReviewer3 · 2019-10-20
**Official Blind Review #3**

**Rating:** 6

**Review:**

The authors derive the influence function of models that are first pre-trained and then fine-tuned. This extends influence functions beyond the standard supervised setting that they have been primarily considered in. To do so, the authors make two methodological contributions: 1) working through the calculus for the pre-training setting and deriving a corresponding efficient algorithm, and 2) adding $L_2$ regularization to approximate the effect of fine-tuning for a limited number of gradient steps.

I believe that these are useful technical contributions that will help to broaden the applicability of influence functions beyond the standard supervised setting. For that reason, I recommend a weak accept. I have some questions and reservations about the current paper:

1) Does pretraining actually help in the MNIST/CIFAR settings considered? These seem to be non-standard pretraining settings. More generally, can we relate influence to some objective measure that we care about (say test accuracy), for example by showing that removing the top X% of influential pretraining data hurts test accuracy as much as predicted? Minor: section 4.2 also seems non-standard. Are the exact same bird vs. frog examples being used for both pretraining and finetuning?

2) In what situations might we want to examine the influence of pretraining data, and can we design experiments that show those situations? For example, perhaps we're wondering if different types of sentences in the one-billion-word dataset might be more or less useful. Can we verify those claims using these multi-stage influence functions? It is otherwise difficult to assess the utility of the qualitative results (e.g., Figure 3 and Appendix C).

3) It'd be helpful to get a better understanding of the technical contributions of this paper. Specifically,
a. What is the impact of $\alpha$ in equation 12 and how does it interact with the number of fine-tuning steps taken?
b. If the Hessian has negative eigenvalues, we can still take $H^{-1}b$ by solving CG with $H^2$, but what does this correspond to? Is the influence equation well defined (or the Taylor approximation justified) if $H$ is not positive definite?



**Experience Assessment:**

I have published in this field for several years.

**Review Assessment: Checking Correctness Of Derivations And Theory:**

I assessed the sensibility of the derivations and theory.

**Review Assessment: Checking Correctness Of Experiments:**

I assessed the sensibility of the experiments.

**Review Assessment: Thoroughness In Paper Reading:**

I read the paper thoroughly.

---

> ### Author Response · Authors · 2019-11-14
> **Thank you for the encouraging comments! We have addressed your questions here and have updated the paper.**
>
> Q1.1: Is the pretrain-finetune reasonable setting? Does pretraining actually help?
> In the experiment, we consider a transfer learning setting where we have lots of samples from source domain (pretraining stage), but fewer data from the target domain (finetuning stage). For this transfer learning setting, since there is not much data from the finetuning task, using the pretrained model from a similar task could help to improve the finetuned model’s accuracy. For example, if we only have 1k examples from 8 classes in CIFAR (8 classes other than bird and frog) to train a model for classifying these 8 classes from scratch, the resulting model can only achieve 49.0% test accuracy. While if we have another 10k training examples from 2 different classes (bird and frog) to pretrain a model and use its CNN layers as our embedding, then finetune the fully connected layers with this fixed embedding on the remaining 8 classes (1k examples), we can achieve 53.5% test accuracy on these 8 classes. So when our finetuning example is limited, pretraining an embedding on similar data actually helps.
>
> Q1.2: Is there any connection between influence score and testing accuracy. If we remove some testing data, how that will impact the testing accuracy.
> There is a strong connection between influence score and loss function, and loss function is related to testing accuracy. As an example, at the pretraining stage, we train a model with examples from only two classes (bird vs. frog) and use the remaining eight classes in CIFAR for finetuning. So the source data is bird vs. frog and the target data is the other 8 classes. After we removed the top 10% highest influence scores (positive values) examples from pretrain (source data), we can improve the accuracy on target data from 58.15% to 58.36%. As a reference, if we randomly remove 10% of pretraining data, the accuracy will drop to 58.08%. Note that when points with positive influence scores are added to the pretraining set, the model’s loss on test is expected to be increased. So removing them from the pretraining set will decrease the loss and improve accuracy.
>
> Q1.3: Section 4.2 also seems non-standard. Are the exact same bird vs. frog examples being used for both pretraining and finetuning?
> We agree that this is not a standard transfer learning setting since the source and target domains are the same. But this section is actually a study on the influence function scores’ relationship with the task similarity. Here we want to show that if the pretraining (source domain) and finetuning (target domain) are similar, the influence scores’ magnitude is expected to be larger. As an extreme case, we let the pretraining and finetuning tasks to be exactly the same.
>
> Q2: In what situations might we want to examine the influence of pretraining data, and can we design experiments that show those situations? Can we verify those claims using these multi-stage influence functions?
> Our model can be used in various situations, for instance, we might want to investigate which pretraining data are highly associated with a test sample that is predicted wrong. After we detect these ‘wrong’ pretraining data, we can remove these data and retrain the model. This could potentially improve model performance. The examples shown in Table B in the appendix are the pretraining examples with the smallest or largest absolute influence function scores. They are identified by the influence function as the least and most useful sentences. In Fig 3, we associate the misclassified test sample with the pretraining data having the highest influence score. Also in the reply to Q1.2, we design a new experiment to show that if we remove 10% of the highest influence scores examples from pretraining data, we can improve the model performance.
>
> Q3.a:  The impact of in Eq (12) and how does it interact with the number of fine-tuning steps?
> In Eq(12), we add $\|W-\bar{W}\|^2_F$ to the finetuning loss, so that 1) finetuned model can utilize the pretrained model’s information. 2) we can build the connection between pretraining and finetuning tasks, otherwise, these two tasks are decoupled, and we could not get the influence score for pretraining data; 3)  when the finetuned model converges, the embedding weights are expected to be close to the pretraining result. It is hard to tell whether Eq(12) would reduce the fine-tune steps as it is a non-convex problem. Also if the $alpha$ goes to infinity,  Eq(12) will be the same as case 1 in Section 3.2.1.
>
> Q3.b: What if the Hessian has negative eigenvalues?
> Since the proof of Thm 1 relies only on Taylor expansion which holds for any $H$, the influence function formulation does hold even if Hessian has negative eigenvalues, as long as it's invertible. If $H$ is invertible and it has negative eigenvalues, we can still run the CG on $H^2x=Hb$ since $H^2$ is PD. What we get is $(H^2)^{-1}Hb$, which is the same as $H^{-1}b$. That is why we use $argmin\  0.5x^TH^2x-b^THx$ in Section 3.3.

---

### Official Review · AnonReviewer2 · 2019-10-23
**Official Blind Review #2**

**Rating:** 3

**Review:**

This is an analysis paper of pretraining with the tool “influence function”. First, the authors calculate the influence score for the models with/without pretraining, and then propose some implementation details (i.e., use CG to estimate the inversed Hessian). To calculate the influence function of a model with pretraining, the authors use an approximation f(w)+||w-w*||, where w* is pretrained.
The experiments are conducted on MNIST and CIFAR.

1.	The idea of converting a pre-trained model with f(w)+||w-w*|| is interesting. But I do not think the conclusion is very promising and convincing. The authors leverage Pearson correlation to measure the similarity between “true loss difference” and “score value”. However, i do not think the value $0.62$ is significant. As shown in Figure (2), intuitively, the linear correlation between these two values do not hold. Also, I am not quite sure about the practical value of calculating influence scores.
2.	The experiments are conduct on small-scale datasets. I am not sure whether the conclusion holds for larger dataset.
3.	Page 7, last paragraph, “we replace all inverse Hessians in (11) with identity matrice”=>why?
4.	In figure 3, what is the relationship between the two MNIST images, and the relationship between the two CIFAR images?


**Experience Assessment:**

I have published one or two papers in this area.

**Review Assessment: Checking Correctness Of Derivations And Theory:**

I assessed the sensibility of the derivations and theory.

**Review Assessment: Checking Correctness Of Experiments:**

I assessed the sensibility of the experiments.

**Review Assessment: Thoroughness In Paper Reading:**

I read the paper at least twice and used my best judgement in assessing the paper.

---

> ### Author Response · Authors · 2019-11-14
> **Thank you for your comments! We have addressed them here and in the revised paper.**
>
> Thank you for your constructive review. We will really appreciate it if you can read our response and provide us some feedback. We will be glad to discuss with you on any further concerns.
>
> Q1.1:  linear correlation between influence score and true loss is not strong
> It is almost impossible to get the exact linear correlation because the influence function is based on the first-order conditions (gradients equal to zero) of the loss function, which may not hold in practice. Therefore people usually report their correlation using Pearson’s r value, e.g., Koh&Liang ICML’17. In Koh&Liang ICML’17, it shows the r value is around 0.8 but their correlation is based on a single model with a single data source, but we consider a much more complex case with two models and two data sources: the relationship between pretraining data and finetuning loss function. So we expect to have a lower r value. In summary, we think 0.6 is reasonable to show a strong correlation between pretraining data’s influence score and finetuning loss difference.
>
> Q1.2: the practical value of calculating influence scores.
> The influence function is to measure how the model performs when removing or adding one training example. It can be used in various ML tasks. A simpler use case is one where we have a bad/undesirable model output and we want to trace that back to the training instances that might have caused that. In the introduction and related work sections, we provide several references for the practical use cases of influence function.
>
> Q2: can we do larger dataset?
> In the appendix, we perform our model on the Elmo model with One-billion-word (OBW) dataset. OBW dataset contains 30 million sentences and 8 million unique words. We give some quantitative results in Table B in the appendix, where we show some test examples and their corresponding largest (in magnitude) influence function score sentences in 1000 randomly selected pretraining sentences in OBW dataset.
>
> It is challenging to get the Pearson’s r value on the Elmo model trained with OBW dataset to show the relationship between influence scores and true loss change. To get Pearson’s r value, we need to remove each example in the pretraining dataset at a time (one sentence in One-billion-word dataset) and then train the model (model with 13.6 million parameters for a small Elmo model) from scratch to see the loss difference before/after removing one training sample. The training for Elmo model on OBW is very time consuming--with 3GPU it takes 14 days. While to get the r value, we need to do that for at least a few hundred pretraining examples. Therefore, while influence scores can be calculated on large datasets such as Elmo, we can only show the r value in the small scale dataset. It is our future work to compute the r value for Elmo model+OBW dataset.
>
> We want to emphasize that the complexity of our method is linear to the number of pretraining examples and the number of parameters. To compute the influence score and get the most influential training samples for Elmo model + OBW data in Table B, our computation is very fast--only taking 20 min to compute influence function scores for 1000 randomly selected pretraining examples. This also explains why influence score is important and useful for ML area as it is usually time-consuming to figure out ‘bad’ training samples by removing one sample and training a model from scratch to check the performance, and influence score provides an analytic way to narrow down the candidate set efficiently.
>
> Q3: Page 7, last paragraph, “we replace all inverse Hessians in (11) with identity matrice”=>why?
> Replacing all inverse Hessions with identity is not our method, but a baseline we compared with as an ablation study. If we replace all inverse Hessians with an identity matrix, the influence scores become the inner product between training data and testing data’s gradient. In Figure 4, this baseline method can only give an r value of 0.17, while for the same setting, our method gets an r value of 0.47 as shown in Fig 2b. This shows that the inverse Hessian is necessary and our method is more accurate for measuring loss change.
>
> Q4: In fig 3, what is the relationship between the two MNIST images, and the relationship between the two CIFAR images?
> In Fig 3, we pair a misclassified test image in the finetuning task with the pretraining image which has the largest positive influence score value with respect to that test image. Intuitively, the identified pretraining image contributes most to the test image’s error. We can indeed see that the identified examples are of low quality, which leads to negative transfer.

---

### Official Review · AnonReviewer1 · 2019-10-29
**Official Blind Review #1**

**Rating:** 6

**Review:**

This paper proposes a multi-stage influence function for transfer learning to identify the impact of source samples to the performance of the learned target model on the target domain. It considers two cases: fixed pretrained parameters and fine-tuned parameters.

Why not to directly add a scaled identity matrix to problem (15) to avoid the non-PSD issue?

How to use the proposed method to identify source samples that cause negative transfer as discussed in the introduction?

Even using the conjugate gradient method to reduce the complexity, the total complexity is still high as the number of parameters in a deep neural network is large. It is better to report the running time to see the efficiency of the proposed method.

In transfer learning, there is a setting that source data are not accessible due to, for example, the purpose of the privacy protection. In this case, can influence function be used?

For presentation, I think it is not correct to use ‘pretrain’ or ‘finetune’ before a noun. They should be replaced with ‘pretrained’ and ‘finetuned’.

**Experience Assessment:**

I have published in this field for several years.

**Review Assessment: Checking Correctness Of Derivations And Theory:**

I assessed the sensibility of the derivations and theory.

**Review Assessment: Checking Correctness Of Experiments:**

I assessed the sensibility of the experiments.

**Review Assessment: Thoroughness In Paper Reading:**

I read the paper at least twice and used my best judgement in assessing the paper.

---

> ### Author Response · Authors · 2019-11-14
> **Thank you for your comments! We have more explanations here.**
>
> Q1: Why not to directly add a scaled identity matrix to problem (15) to avoid the non-PSD issue?
> We modified the text below Eq(15) in the paper. (15) is to get $H^{-1}b$, no matter $H$ has negative eigenvalues or not. To use CG, we can solve either $argmin\ 0.5x^THx-b^Tx$ or $argmin\ 0.5x^TH^2x-b^THx$. For both we need $H$ or $H^2$ to be PD. $H^2$ is always PD as long as $H$ is invertible. If $H^2$ is not ill-conditioned, we solve the second formulation directly without any further modifications. If $H^2$ is ill-conditioned, we add a damping term $\lambda I$ to $H^2$ where $\lambda$ is very small for numerical stability and to avoid ill-condition as explained in the paper. While if we solve $argmin\ \frac{1}{2}x^THx-b^Tx$ using CG and $H$ is not PD, we always need to add a $\beta I$ to make $H+\beta I$ PD and $\beta$ should be larger than the absolute value of $H$’s smallest negative eigenvalue. When $\beta$ is large, the solution of $argmin\ 0.5x^T(H+\beta I)x-b^Tx$  can be very different from $H^{-1}b$.
>
> Q2: how to connect the influence function with negative transfer examples?
> The pretraining examples with large positive influences scores are the ones that will increase the loss function value indicating negative transfer. Based on the influence score, we could improve the negative transfer issue. For example, at pretraining stage, we train a model with examples from 2 classes (“bird" vs. “frog") and use the remaining 8 classes in CIFAR for finetuning. So the source data is “bird" and “frog" and the target data is the other 8 classes. After we remove the top 10% highest influence scores examples from pretraining data, we can improve the test accuracy on target data from 58.15% to 58.36%. These 10% highest influence scores training samples are negative transfer examples. As a reference, if we randomly remove 10% of pretraining data, the accuracy will drop to 58.08%.
>
> Q3: time complexity issue and training time results.
> At the end of Section 3, we discuss the time complexity of computing the inverse Hessian vector product (IHVP). As explained in Section 3.3, we do not compute or store Hessian explicitly but only compute IHVP. All the computation related to inverse Hessian can use IHVP, which makes the computation efficient.
>
> If the pretrained and finetuned model have $p_1$ and $p_2$ parameters, respectively, and we are given $m$ pretraining examples and $n$ finetuning examples. The time complexity for the 2 IHVPs are $O(m*p_1*r)$ and $O(n*p_2*r)$, where $r$ is the number of CG iterations. The total time complexity of computing a multi-stage influence score for all the training data is thus $O((n*p_2+m*p_1)*r)$--linear to number of training samples and model parameters. In practice, the influence function computation is fast. For example, on the CIFAR dataset, the time for computing influence function w.r.t all pretraining data is 230s (with roughly 200 iterations of CG for 2 IHVPs in Eq. (10) and Eq. (11) ). Also in appendix B we run an ELMo model with 13.6 million parameters. We compute the influence score for this large model and get the most influential pretraining examples in OBW data in Table B. Our computation is very fast--only takes 20 mins to compute influence function scores for 1000 randomly selected pretraining examples.
>
> Q4: can influence function be used for the case when the source data is not available?
> What we need for computing the influence function from the source data is the gradient. So our method can be used even when the source data itself is absent, but a black box unit is given to provide the gradient of each source data to the influence score computation in Eq. (11). Note in Eq. (11), the terms in the bracket are all gradients/Hessians of the finetuned model. They do not depend on the pretrained model or the pretraining data.
>
> As a concrete example, company A pretrains a model on its own private source data and provides only the pretrained model to company B for a downstream task. If B does not think the pretrained model is good enough for its downstream task, B can compute the terms in the bracket of Eq. (11) to get a vector, which is $\frac{\partial f}{\partial W}$. Then B can send this vector to A, without sharing its downstream task’s model or data. A can calculate the inner product of this vector with $I_{z,W}$ to compute the influence function score in its private pretraining (source) data to identify the examples’ contribution to the downstream task’s error. In this scenario, the multi-stage influence function score can be obtained without any exchange of model or data. So if B does not have access to the source data, it can ask A (who has access) to compute influence score and debug the embedding for it, by just sending a vector $\frac{\partial f}{\partial W}$ to A.
>
> Q5 It is not correct to use ‘pretrain’ or ‘finetune’ before a noun. They should be replaced with ‘pretrained’ and ‘finetuned’.
> Thanks for the suggestion. We have made the changes in the revised version.

---

### Author Response · Authors · 2019-11-15
**Summary of rebuttal**

We thank all the reviewers for their constructive comments! Below is a summary of our reply to the reviewers and the main changes to the paper.

1) To answer R1 and R3’s questions on the use case of our model, we ran additional experiments to show that removing examples from the highest influence scores in pretraining data can be used to improve finetuned model’s test accuracy.

2) To address R1 and R2’s comment on time complexity, we report the running time of our method on both large ELMo and small CIFAR models to show its scalability.

3) For R2’s concern on r value, we explain why we think Pearson’s r value of 0.6 is meaningful to show the strong correlation given the complexity of this task.

4) To address R3’s question on the experiment setting, we have new experiments showing that pretrained embedding does improve the model’s accuracy when finetuning examples are limited.

5) We have answered the questions related to CG and Hessian matrix to be non-PSD from R1 and R3 in the reply, and modified Section 3.3 in the new version of the paper to make things clear.

6) We have fixed the typos as pointed out by R1 in the revised paper to make it easier to follow.

We hope the above changes and replies can address the questions/concerns from the reviewers for this paper.

Thanks,
Paper2011 Authors

---

### Decision · Program_Chairs · 2019-12-19

**Decision:**

Reject

**Comment:**

This paper extends the idea of influence functions (aka the implicit function theorem) to multi-stage training pipelines, and also adds an L2 penalty to approximate the effect of training for a limited number of iterations.

I think this paper is borderline.  I also think that R3 had the best take and questions on this paper.

Pros:
 - The main idea makes sense, and could be used to understand real training pipelines better.
 - The experiments, while mostly small-scale, answer most of the immediate questions about this model.

Cons:
 - The paper still isn't all that polished.  E.g. on page 4: "Algorithm 1 shows how to compute the influence score in (11). The pseudocode for computing the influence function in (11) is shown in Algorithm 1"
 - I wish the image dataset experiments had been done with larger images and models.

Ultimately, the straightforwardness of the extension and the relative niche applications mean that although the main idea is sound, the quality and the overall impact of this paper don't quite meet the bar.